# Complexity Analysis of the Default Mode Network Using Resting-State fMRI in Down Syndrome: Relationships Highlighted by a Neuropsychological Assessment

**DOI:** 10.3390/brainsci11030311

**Published:** 2021-03-02

**Authors:** María Dolores Figueroa-Jimenez, María Carbó-Carreté, Cristina Cañete-Massé, Daniel Zarabozo-Hurtado, Maribel Peró-Cebollero, José Guadalupe Salazar-Estrada, Joan Guàrdia-Olmos

**Affiliations:** 1Department of Health Sciences of the Centro Universitario de los Valles, University of Guadalajara (México), 44160 Guadalajara, Mexico; maria.figueroa@academicos.udg.mx (M.D.F.-J.); jsalazar@valles.udg.mx (J.G.S.-E.); 2Institute of Neuroscience, University of Barcelona, 08035 Barcelona, Spain; mcarbo@ub.edu (M.C.-C.); mpero@ub.edu (M.P.-C.); 3Serra Hunter Fellow, Department of Cognition, Developmental Psychology and Education, Faculty of Psychology, University of Barcelona, 08035 Barcelona, Spain; 4Department of Social Psychology & Quantitative Psychology, Faculty of Psychology, University of Barcelona, 08035 Barcelona, Spain; cristinacanete@ub.edu; 5UB Institute of Complex Systems, University of Barcelona, 08028 Barcelona, Spain; 6RIO Group Clinical Laboratory, Center for Research in Advanced Functional Neuro-Diagnosis CINDFA, 44160 Guadalajara, Mexico; daniel.zarabozo@gmail.com

**Keywords:** DMN, down syndrome, fMRI, IQ, resting state, neuropsychology

## Abstract

Background: Studies on complexity indicators in the field of functional connectivity derived from resting-state fMRI (rs-fMRI) in Down syndrome (DS) samples and their possible relationship with cognitive functioning variables are rare. We analyze how some complexity indicators estimated in the subareas that constitute the default mode network (DMN) might be predictors of the neuropsychological outcomes evaluating Intelligence Quotient (IQ) and cognitive performance in persons with DS. Methods: Twenty-two DS people were assessed with the Kaufman Brief Test of Intelligence (KBIT) and Frontal Assessment Battery (FAB) tests, and fMRI signals were recorded in a resting state over a six-minute period. In addition, 22 controls, matched by age and sex, were evaluated with the same rs-fMRI procedure. Results: There was a significant difference in complexity indicators between groups: the control group showed less complexity than the DS group. Moreover, the DS group showed more variance in the complexity indicator distributions than the control group. In the DS group, significant and negative relationships were found between some of the complexity indicators in some of the DMN networks and the cognitive performance scores. Conclusions: The DS group is characterized by more complex DMN networks and exhibits an inverse relationship between complexity and cognitive performance based on the negative parameter estimates.

## 1. Introduction

Down syndrome (DS) is one of the most frequent diagnoses in the intellectual disability field and is characterized by a specific cognitive phenotype due to alterations in hippocampal structure [1]. The neuropsychological DS profile is characterized by alterations in motor abilities, language (morphosyntax), verbal short-term memory and explicit long-term memory; in contrast, visuospatial short-term memory and implicit long-term memory are relatively preserved [2,3]. These patterns in DS were described in a recent systematic review [4]. In this review, the authors concluded that these dysfunctions were related to chronic health conditions, basically sleep disorders. In addition to these sleep disorders present in DS, another important issue addressed in the DS population related to neuropsychological aspects is Alzheimer’s disease [5,6,7]. Although the presence of dementia in this population becomes apparent at approximately the fifth decade of life [8], the indicators of Alzheimer’s disease are expected to be present ten years before, that is, at approximately 40 years old [9].

Interest in neuropsychological and cognitive functions in people with DS has promoted the development of assessment tools specifically designed for this population, such as the TESDAD battery [10] and the Arizona Cognitive Test Battery (ACTB) [11,12]. Moreover, in recent years, there has been a growing interest in introducing neuroimaging techniques to study neuropsychological aspects in people with DS, mainly in studies related to Alzheimer’s disease and dementia [13,14].

In relation to neuroimaging techniques, the use of functional magnetic resonance imaging (fMRI) is increasingly used to study brain connectivity in people with intellectual disability [15]. The use of fMRI is becoming more common in brain connectivity studies, because it allows the understanding and analysis of the brain as a complex network [16,17,18]. From this perspective, the brain is understood as a system organized by nodes and the connections established between them that provide functional or structural information [19]. This discipline has its origins in mathematical graph theory with the aim of quantifying and defining how the brain is organized [20]. There are two main groups of measures to study brain networks: first, indicators of functional segregation that define local network communities; second, measures addressed to assess functional integration. These latter indicators explain the global communication between the segregated groups, that is, how these networks are coordinated and share information [21,22].

Graph theory has been used in studies of resting-state fMRI (rs-fMRI) to provide a better comprehension of brain connectivity in neurological or psychiatric diseases [23,24,25]. In rs-fMRI, when the person is required to maintain closed eyes or look at fixed points, a specific network of the brain is activated, called the default mode network (DMN) [26,27]. This network is characterized by high levels of functional and structural connectivity and by high levels of resting metabolic activity in healthy people [27]. A recent study [28] evaluated DMN properties in young DS people in comparison with a control group. The results showed that there were higher levels of overactivation in the ventral, sensorimotor and visual DMN networks, although these effects occurred with high levels of heterogeneity in connectivity patterns. However, it is not clear whether this heterogeneity was due to cognitive performance variables.

In the present study, the principal goal was to provide knowledge regarding DMN function in the DS population and analyze whether complexity in the brain network could be related to cognitive performance. A complex system is defined by the study of phenomena in which multiple sources of information are offered, focusing the analysis on all sources of information regardless of the operation of a single source of information. A complex system, therefore, requires simultaneous and multiple information at a certain point in time. There is no strict definition of this concept, since its application to various scientific fields makes it difficult. However, there is a wide degree of agreement in considering the quantification of complexity as indicators of (1) How hard is it to describe? (2) How hard is it to create? and (3) What is its degree of organization? [29].

In DS, DMN connectivity to other areas of the brain is different from that in the non-DS population [30,31,32]. Moreover, based on graph theory analysis, the clustering coefficient was higher in the DS group than in the control group [30]. Nevertheless, Carbó-Carreté et al. [15], in a systematic review to assess brain activity in people with DS, identified that there is no typical, regular and stably established functional connectivity. In addition to the studies on brain connectivity in people with DS, the published research regarding AD (Alzheimer Disease) has become a framework for our work because, as mentioned, it is one of the most important issues being addressed in this population based on their early appearance and types of neuropsychological alterations presented. Based on AD and graph theory studies [33], it was shown that the characteristic path length (functional integration measure) and the clustering coefficient (functional segregation measure) were low. Thus, the small world, defined by the optimal balance between these two measures [34], indicated altered brain connectivity. Nevertheless, a recent study [35] examined the topological attributes of the small world in patients with subjective cognitive decline (SCD; a clinical stage before the diagnosis of AD). The SCD sample presented global efficiency values lower than normal controls, while characteristic path length and modularity were higher. However, increased characteristic path length values and decreased global efficiency suggested a reduction in abilities related to information communication of the whole brain (i.e., integration function) in the SCD sample. However, the increased values of modularity in SCD suggested that, in this population, local communication, and information transmission (segregation function) were improved.

Some studies in healthy populations have also presented results that provide interesting data to define our objectives. For instance, it was shown in healthy older people (<75 years) that there was less connectivity within the network but more connectivity among the subnetworks that make up the DMN [36]. They used indicators of lower segregation, modularity and local efficiency. The results obtained were associated with poor executive function, memory and processing speed [37]. All this reflects a progressive loss of specialization within the brain networks related to higher functions [38].

So far, the cited studies do not allow us to think of a very stable structure in the behavior of these indicators in a healthy or clinical population. In general terms, it can be expected that connectivity networks in healthy populations present values in these indicators that are associated with a very stable network structure and with a high density of connections, although with not particularly high connection values. Therefore, with low values in the number of communities, in the mean and standard deviation of path length, of Clustering or of Small World. In the same way, we can find higher values in the Density, Complexity or Number of Triangles indicators in the same population.

Based on the published works mentioned, the aim of this paper was to explore the observed distribution of complexity indicators in a DS group in comparison with a matched control group. Our expectation is that some of the descriptive measures of the connectivity networks show more erratic observed distributions in the group of people with DS. Moreover, we analyzed the relationship between some of those complexity indicators in the subareas that constitute the DMN and the scores obtained on neuropsychological tests. Given the dispersion that we expect to find in the group of people with DS, the relationships with neuropsychological performances will be specific and not systematic.

## 2. Materials and Methods

### 2.1. Participants

The sampling was non-random and was performed through contact with different associations dedicated to DS in the State of Jalisco (México) (53.2% of participants) and Barcelona, Spain (46.8%). The initial sample comprised a total of 32 persons with DS between the ages of 16 and 35 (M = 24.7 and SD = 5.49), of whom 28.12% were women (number of women = 9). The inclusion criteria applied were (a) age between 16 and 35 years old and (b) a diagnosis of DS. The exclusion criteria were (a) evidence of other diagnostic comorbidities involving cognitive dysfunction with AD; (b) inability to obtain legal consent from guardians; and (c) the presence of medication affecting cognitive function.

The diagnosis proportions of intellectual disability of the 29 participants with DS referred by the tutors (the remaining could not be accredited) were 3.4% borderline intellectual disability, 52.2% mild intellectual disability, 37.9% moderate intellectual disability and 3.4% profound intellectual disability. This classification appeared in the official report that each DS person presented at the time of incorporation into the study, and limited intellectual disability relates to the borderline zone, so this category does not appear in the ICD-10 categories (Codes F70-F79). A total of 84.4% of the participants with DS were right-handed, and 6.3% of the participants with DS were ambidextrous (*n* = 32).

Written informed consent was obtained from every individual before taking part in the study in accordance with the Declaration of Helsinki and with the approval of the institutional ethics committee. Moreover, this procedure was approved by the Bioethics Committee of the University of Barcelona (03/10/2017).

Regarding the fMRI signal recording, ten participants with DS recorded an excess of movement greater than ±2 degrees (or greater than half a voxel size) and were eliminated, and some of them were eliminated even repeating the recording session, since the second session also showed excess movement. The final sample was ultimately composed of a total of 22 persons with DS, with an observed age distribution of M = 25.55 and a standard deviation (SD) = 5.119. The distribution of the sample by sex was 22.7% female. In the final sample, the maximum movement was 1.2 degrees, and the average was 0.72 degrees (SD = 0.11).

A control group was added, matched (one by one) by age and sex with the DS group. Subjects were selected who were within the range of movement used in the group of people with DS, movements of a maximum of ±2 degrees (M = 0.92; SD = 0.09) and only subjects whose protocol contained the absolute absence of pathology were included that compromised their cognitive performance or any type of chronic disease or medication. The images of this second sample were obtained from the Connectome Project (http://www.humanconnectomeproject.org/) during July 2020 with the same image properties as those in the DS group. For each control participant, we obtained structural T1 and T2 images and whole-brain resting-state fMRI signals during the same period as the DS group. In all the individuals in the control group, the number of volumes was greater than in the DS group (between 240 and 300 volumes). Therefore, we used only the first 220 volumes corresponding to those used in the DS group.

### 2.2. Instruments

The DS data from this work are part of a larger protocol in which the relationship between brain signals (fMRI) and various variables connected with cognitive performance, quality of life and physical activity are studied. In all cases, the following elements of assessment and measurement were administered to determine if they met the criteria for inclusion and exclusion:(1)Ad hoc questionnaires were used to assess the clinical and educational history, and the following variables were collected: age, sex, place of residence and degree of intellectual disability.(2)Dementia Screening Questionnaire for Individuals with Intellectual Disabilities (DSQIID): with an internal consistency estimated with Cronbach’s α of 0.91 [39]. This questionnaire was useful for ruling out signs of dementia. As it only affected the application of the exclusion criteria, a version adapted to Spanish was used without a study of its psychometric properties.

Neuropsychological evaluation:

The protocol designed for the evaluation of the participants with DS to measure cognitive performance was integrated with the following neuropsychological tests:Frontal Assessment Battery (FAB): This consisted of tasks exploring the functions of the frontal lobes through six subtests: similarities (concept formation), verbal fluidity (mental flexibility), motor series (programming), interference (carrying out conflicting instructions), control (inhibition of responses) and autonomy (independence from the external environment). The cutoff point for frontal-subcortical deficits was 16–15, and the cutoff point for frontal-subcortical dementia was 13–12. The Frontal Assessment Battery scores showed a correlation with the Mattis Dementia Rating Scale scores (rho = 0.82, *p* < 0.01) [40].Intelligence Quotient (IQ) was assessed using the Kaufman Brief Test of Intelligence (KBIT), a screening test that evaluates crystallized intelligence (learning and problem solving) based on formal schooling and cultural experience, from two levels of conceptualization: verbal intelligence with an expressive vocabulary and definitions subtest and nonverbal intelligence with a master’s subtest. This test is valid for use by people from 4 to 90 years of age and generates standard scores (verbal, nonverbal and IQ composite) [41].

Regarding the control group, only the rs-fMRI image was analyzed. These data are open and regulated by Connectome Project regulations. The cognitive outcome assessment in the control group was different from that in the DS group and impossible to compare. Therefore, to avoid confusing comparisons, the cognitive assessment in the control group was excluded. The inclusion criterion in the control group required that the technical characteristics of the acquisition of the images were the same as those used in the registration of the people in the DS group. This was done to guarantee the comparison of structures of the connectivity networks between the two groups.

### 2.3. Procedure

Each participant with DS and their guardians provided informed consent before the first neuropsychological evaluation session in accordance with the Declaration of Helsinki. The protocol was approved by the Bioethics Commission of the University of Barcelona. Additionally, a medical report was obtained from each participant to confirm that the MRI study was safe. All participants were evaluated in two registration sessions by previously trained researchers. The administration sequence was the same for all participants, and the previously referenced scales were administered first to avoid fatigue bias. All the questionnaires were administered by the researcher. The sociodemographic information was obtained from the people with DS, and all of this information was collected on the same day. The DSQIID scale was completed by the guardians of the participants with DS.

### 2.4. MRI Acquisition and Preprocessing

After administering the scales, the participants in the DS group had the fMRI recording sequences performed in the following order: T1, T2, FLAIR and 6-minute resting state. Both Mexican and Spanish participants were recorded on similar scanners. Two Philips Ingenia 3.0T system models were used (one located at the Clinical Laboratory, Integral Centre of Medical Diagnosis of Guadalajara’s Grupo Río in Jalisco, and the other at the Fundació Pasqual Maragall in Barcelona). A T1-weighted turbo field echo (TFE) structural image was obtained for each participant with a 3-dimensional protocol (repetition time (TR) = 2300 ms, echo time (TE) = 2980 ms, 240 slices, and field of view (FOV) = 240 × 240 × 170). The image acquisition was in the sagittal plane. For the functional images, a T2 weighted (BOLD) image was obtained (TR = 2000 ms, TE = 30 ms, FOV = 230 × 230 × 160, voxel size = 3 × 3 × 3 mm, 29 slices). The image acquisition was in the transverse plane. The characteristics of both scanners were identical, and a subsequent review of each recording was performed to check if there was any difference between the two recording facilities. No difference was found between the two, neither technologically nor procedurally. To guarantee the equality of records in both scanners, data from a reduced group of subjects was recorded to determine if there was any significant difference between the records of the same person in the two scanners. This procedure was performed prior to this work and did not show any relevant difference between scanners. During scanning, the participants were instructed to relax, remain awake, and keep their eyes open and fixed on a cross symbol on the screen. The data were collected during the period from March 2018 to July 2019.

In the case of the control group, the acquisition was performed in different institutions in the United States. The repetition time (TR) in all cases was 2000 ms, and the voxel size was different for every protocol. As mentioned above, the technical characteristics of both groups were the same and only open-eye resting-state protocols were selected.

For the two groups, the structural imaging data were analyzed using an FMRIB Software Library (FSL) [42] preprocessing pipeline adapted under authorization from Diez et al. [43], with its parameters adjusted to fit our experimental data, including a motion correction procedure to solve the undesired head movements in the fMRI sessions. To obtaining the functional connectivity (FC) matrices, the fMRI images were preprocessed as follows. First, a slice time correction based on the TR of the image acquisition was carried out to obtain thirty contiguous slices in the Anterior commissure–posterior commissure (AC–PC) plane. The input images were reoriented to match the template axes and motion correction was computed to coregister all the volumes with the central one so that all the voxels of the different volumes belonged to the same brain point. Then, all non-brain tissue was removed and, to get a better signal-to-noise ratio, the volumes were smoothed with a 6 mm full width at half minimum (FWHM) isotropic Gaussian kernel. Also, intensity correction and band-pass filtering between 0.01 and 0.08 Hz were applied to the data. The resulting functional data images were registered and normalized to the standard Montreal Neurological Institute (MNI) space. Finally, the white matter and the cerebrospinal fluid effects were removed so that no other interference was added to the fMRI signal. The final step involved registering our structural data images to the normalized space using the Montreal Neurological Institute (MNI) reference brain based on the Talairach and Tournoux coordinate system [44]. 

### 2.5. Regions of Interest

The automated anatomical labeling atlas 90 (AAL90) [45] was used to define the regions of interest (ROIs). This atlas contains 45 cortical and subcortical areas in each hemisphere (90 areas in total), which are alternatively interspersed (available by request) and described in Table 1. To acquire the full signal from a given ROI, it is necessary to compute an average over the entire time series of all the voxels of a given brain area following the AAL atlas. The specific values of each ROI were estimated from the application of Principal Components Analysis (PCA) with the strict selection of each group of voxels defined by the mask of each ROI. Given the objective of the present study regarding brain connectivity patterns, we identified only the DMN. These regions were divided into five subnetworks: concentrated partial DMN, anterior, ventral, and two posterior subnetworks, sensorimotor and visual, all based on the classification proposed by Huang et al. [46].

### 2.6. Estimation of Mental Age in the DS Group

The estimation of mental age in the DS population has been widely studied and is also very controversial because of both the absence of specific instruments for this purpose and the use of Intelligence Quotient (IQ) to define the degree of intellectual disability [47,48]. Since it is a measure that involves the recording of cognitive performance, it can mask specific deficits due to its very heterogeneous nature.

In recent years, there has been a growing interest in facilitating this calculation, since, in the latest version of the world reference test for the estimation of IQ in children (WISC-V), the planning of the calculation of mental age in an easy and uncomplicated way has been incorporated. However, in the situation in which we find ourselves, with a sample of persons with intellectual disabilities, there are many difficulties in finding a valid and reliable test for the estimation of mental age [11,49]. Among these difficulties are floor effects, tests that focus on evaluating only language skills, low sensitivity of the measures to detect some effects, low flexibility for use across cultures and languages, applications in a chronological age range that do not directly lead to adjustments for mental age and, finally, lack of psychometric validation in populations with developmental disabilities [11].

Initially, the use of the WISC was proposed for the estimation of IQ in the adult population with DS. The aim was to alleviate the floor effect and hope that an IQ value of approximately 70 points could represent a large part of the population. However, despite the facility of the WISC-V test for the evaluation of the subscales involved in the mental calculation tasks, it does not appear to be suitable for the integral evaluation of persons with ID for various reasons, but there are two reasons in particular: (1) there is a difficulty in understanding items with high verbal content, and (2) it is a test with a very limited age range (between 5 and 16 years). Therefore, we believe that the estimation of mental age should be approached from another perspective [50].

In a study by Hamburg et al. [49], a systematic review of the literature on the different IQ tests for adults with DS was conducted, and of all of them, the one identified with the lowest problem involving a floor effect was the KBIT test, even for very extreme populations (e.g., with dementia). Therefore, the KBIT test was chosen because it extends the range of chronological ages and because it involves less time for application, as there are only two subscales: verbal intelligence (with the expressive vocabulary and definitions subtest) and nonverbal intelligence (with the matrix subtest), which falls within the estimated range of concentration time (approximately 30 min) and reduces the fatigue of the person evaluated.

However, this test does not allow the estimation of mental age. Our proposal was based on the following heuristic. First, direct scores were calculated for the two subtests (matrices and vocabulary), and the IQ of each participant with DS was calculated traditionally. Based on these direct scores, we located in the standardization scales to which age range this score would correspond, selecting the scale with the smallest difference between the population mean (IQ = 100) indirect score and the observed score. Once the scale with the smallest difference was located, the observed score was placed, and the age range to which it corresponded was identified. In all cases, the most favorable mental age (upper limit) within the range offered by the scales was selected. In the case that the direct score indicated a level lower than four years, the mental age was set at that age, assuming that this value is the floor of the test.

The limitations of this proposal are obvious. First, it would be ideal to have a test where the lower limit was less than four years and, second, there is an overestimation effect of mental age because the upper limit of the age confidence interval was used. However, this allows us to avoid the bias effects that could occur in subsequent statistical analyses by facilitating the incorporation of mental age as a relevant variable.

It should be noted that, even with these limitations, we believe that this is one of the most reliable tests for people with DS [49], which is supported by its regular use as an inclusion criterion [31,51,52,53,54].

### 2.7. Statistical Analysis

For the two groups, the observed distributions of the indicators of the nondirected connectivity networks were analyzed based on the ROIs of the five networks described in Table 1, as well as the global network that would freely incorporate the 48 ROIs defined. For each of the five smaller networks and the global network, the nondirected networks were estimated based on the partial correlations between nodes (ROIs), and the complexity indicators are shown in Table 2.

The selection of these indicators has followed two criteria: (1) the use of the most common and known indicators according to Rubinov [21] and (2) previous recommendations [55] regarding sensitive indicators for the description of a network. There are a multitude of possible alternative indicators, but neither the sample size nor the objectives of this paper focus on a comprehensive analysis.

In the DS group, each of the estimates of these indicators for each subarea was included as a predictor for the cognitive performance variables (FAB total score) and the standardized scores of the scale (vocabulary and matrices) KBIT in a specific study, as mentioned above, with the DS group.

For each criterion variable, the resulting multiple regression linear model was obtained from the best possible combination of linear predictors. The following statistical operations were performed on each model. Given the high variance in some of the predictor variables, a 5% cutoff for each tail was used to avoid the effect of extraneous values in such a small sample, and chronological age was included as a correction criterion in the estimation. Once these transformations were made, robust stepwise regression models were estimated using as an inclusion criterion the significance of the change in the coefficient of determination (*R*^2^) and the adjustment value of the Akaike information criteria (AIC). The detection criterion of the best model was a significant change in *R*^2^ with *p* < 0.01 and a more than 10% reduction in the AIC value between successive steps.

## 3. Results

Table 3 shows the basic statistics of the observed distribution of the criterion variables for the DS group. Table 4 shows the complexity indicator statistics for both groups described in Table 2. Estimates of the standard error of the mean were obtained by bootstrapping with 10,000 repetitions to reduce the effect of the small sample. Despite the small sample size, we believe that it is important to observe the behavior of the indicators described to establish the individual and group differences concerning the connectivity networks studied.

To analyze the differences between the two groups, we chose the Mann–Whitney nonparametric test to avoid the effect of the reduced sample size and some anomalies observed in the symmetry and kurtosis of the distributions. The global and subarea results indicated that the median values in the DS group were systematically higher than those in the control group. The unilateral significance of these statistical contrasts ranged from *p* < 0.05 to *p* < 0.001 using the Bonferroni correction to reduce the probability of making a type I error. These results indicated that the overall complexity levels of the fMRI connectivity networks were much higher in the DS group than in the control group.

Likewise, the variance values of the complexity indicators for both groups were compared to assess the hypothesis of greater variability in the DS group. For this, the Levene test adapted to two groups was used, obtaining a clear significance (ranging from *p* < 0.05 to *p* < 0.001) showing that the variability in the complexity indicators in the DS group was greater than that in the control group.

It was only in the DS group that the linear models relating the neuropsychological test performance with the complexity indicator values of each analyzed network were estimated, and these included the estimation for two mental ages. Table 5 shows the results of the weighted least squares (WLS) estimation to reduce the impact of small samples and the Akaike information criterion (AIC) for each model. Table 5 shows the Pearson correlation between each predicted variable (FAB total score, vocabulary subtest score and matrices subtest score) and the complexity indices as regressors.

Table 6 shows the estimated model significance using WLS. To facilitate the statistical estimation processes, only those complexity indicators with statistically significant correlations with each of the three criterion variables were defined as regressors. In this way, some collinearity problems derived from an excessive number of regressors were reduced. Figure 1 shows the different plots representing the significant effects mentioned in Table 6.

## 4. Discussion

In the present study, we compared the complexity of the DMN (using common indicators of complexity [21] in the subareas that constituted the 48 ROI extended network) (Table 1) in the rs-fMRI paradigm between the DS group and a control group matched by sex and age. The results showed that the DS group had more complex networks in the different subnetworks identified as the DMN. Moreover, the observed distribution of those complexity indicators in the DS group revealed greater variability than that in the control group.

Focusing on the network complexity measures, the results obtained are similar to those presented in previous studies. In relation to segregation measures, the values were higher in the persons with DS than in the controls in each subnetwork of the DMN. For instance, consistent with [29], the DS group showed higher global clustering coefficient values than the control group.

In relation to the results provided by studies on AD and people with SCD, the integration measures indicated some abnormalities. The increased values in the mean of the weighted path were similar to the results presented in [35]; thus, we can also argue that in DS people, the capacity to transmit and connect the information through the whole brain is altered. Nevertheless, related to [35] and specifically in their results on modularity, we were also able to affirm that the network segregation function is increased in people with lower cognitive performance. In the current study, this increase in the DS population can be identified through measures of the global clustering coefficients, complexity, and number of triangles. As expected, and reflected in Xu et al., density was higher in subnetworks where the global clustering coefficient was higher, specifically in the DMN partial network and in the DMN anterior network.

In the DS group, we must highlight the peculiar behavior of the observed distribution for all the variables since most of them were characterized by nonsymmetric distributions. Based on the sample size used, these results are no more than initial descriptions. However, the differences in the estimation of the DMN subnetworks in people with DS are very large. The mean value in the number of communities in each subnetwork indicated the impossibility of assuming that each subnetwork was configured as a single directed network. This should be interpreted as an anomaly in the structure of the global DMN and its networks in terms of network complexity. It is sufficient to observe the average value of the number of communities in the global network (48 ROIs) (DS group: M = 4.86 SD = 1.20; and control group M = 1.184 SD = 0.252; *p* < 0.001) and see how far it is from what is expected in the DMN [27]. 

All these results confirm the appraisals regarding the aberrant behavior of the connectivity networks in people with DS. This aberrant behavior can be summarized by saying that the networks show many connections between the ROIs, of a highly disaggregated nature and with a wide within-subject variability, so that it is feasible to think that this type of connectivity pattern may be associated with serious cognitive alterations. This statement is based on the studies cited above that obtained similar results to ours in samples of people with AD. This level of congruence allows us to think about a certain pattern typical of the functional connectivity networks associated with severe cognitive deficit that should be verified with greater force with other interest groups and cognitive evaluation systems.

In addition, we also analyzed the relationship between some of those indicators and the scores obtained in neuropsychological tests for the assessment of executive functions and IQ in the DS group only. Statistically significant results were found in the prediction of the FAB test scores and the vocabulary and matrices subtests of the KBIT test. The significant parameters indicated a positive effect of mental age derived from the KBIT (vocabulary) scores and the negative effect of some complexity indicators.

Regarding the complexity indicators as predictors of some cognitive performance and IQ tests, our results, in general, showed little effect. Moreover, the high variability in the indicators hinders identification of possible effects. In our sample, there were a limited number of indicators that showed a statistically significant impact on the prediction of FAB or KBIT scores. Regarding the effects associated with the mental age variable, they must be interpreted within the logic of the expected effects and consistent with the methodology used for their estimation, which was described above. This supports some previous proposals that can be consulted in [15] in the sense of controlling age (chronological and mental) across groups and the definition of ID in the context of group designs (paired or not).

Another important effect was the appearance of the number of communities in some networks as a significant effect in the prediction of KBIT matrix scores. First, the signs of the parameters allow us to verify our expectation that increases in complexity are related to worse performance in vocabulary and executive functioning. Second, one should consider the possibility that these network changes could also be an indicator related to a low cognitive level of performance. That is, a higher number of communities implies a greater network disaggregation, which would indicate greater complexity and, therefore, is associated with a low psychometric score.

In the FAB and vocabulary scores (KBIT) predictions, this effect was also enhanced. The negative-weighted mean path length in the sensorimotor network, the visual network small-world network, and the complexity value of the ventral DMN indicated the same trend and interpretation. Despite the high variance in the data and the peculiar behavior of the variables, clipped distributions of some statistically significant effects were observed, which allows us to conclude that the complexity structure of the functional connectivity networks is inversely related to cognitive performance, specifically to vocabulary and executive functioning.

These results can be congruent with some others in different populations with psychological, psychiatric and neurobiological disorders: not in the sense of estimating the complexity of nondirected networks but in the sense of studying abnormalities in the definition of directed networks [56,57,58,59,60,61]. 

Regarding the limitations of the study, we identified those that were insurmountable, which are summarized below. The use of the KBIT for the cognitive evaluation of people with DS has clear limitations, although it has the merit of being a short and simple test for a complex evaluation of these characteristics. Another aspect is the mental age estimation approach to which is only an initial proposal and has an unavoidable floor effect.

Finally, the small sample size leads us to interpret our results with caution, although the sample size is larger than that used in most published papers to date. Even so, the size should be improved, and a control group matched for the mental age should be considered. Any registration of participants has not been possible in the last few months. Nevertheless, the results obtained allow us to justify the necessity to explore in more detail the behavior of each subnetwork in the DS population. Most likely, subsequent studies should follow the analysis of recently published articles [35,62], which include studies on the topology of brain networks using the same (or similar) segregation and integration measures used in this paper.

## 5. Conclusions

To summarize the main conclusions, in our opinion, we can highlight that the current paper is the first study that has been conducted to determine the behavior of DMN subnetworks through nondirected networks in people with DS.

Our results indicated that the complexity in the structure of the DMN and the analyzed subnetworks was higher in the DS group than in the control group.

There was enormous variability between participants regarding the network’s behavior.

Some indicators of complexity in the DS group (i.e., path length, complexity and small-worldness) had statistically significant and negative impacts on the prediction of performance for some neuropsychological tests—in this case, the FAB and KBIT.

These results are congruent with the behavior of other segregation or integration measures studied in other populations in which inverse relationships have also been evidenced with other types of psychometric indicators of cognitive performance.

## Figures and Tables

**Figure 1 brainsci-11-00311-f001:**
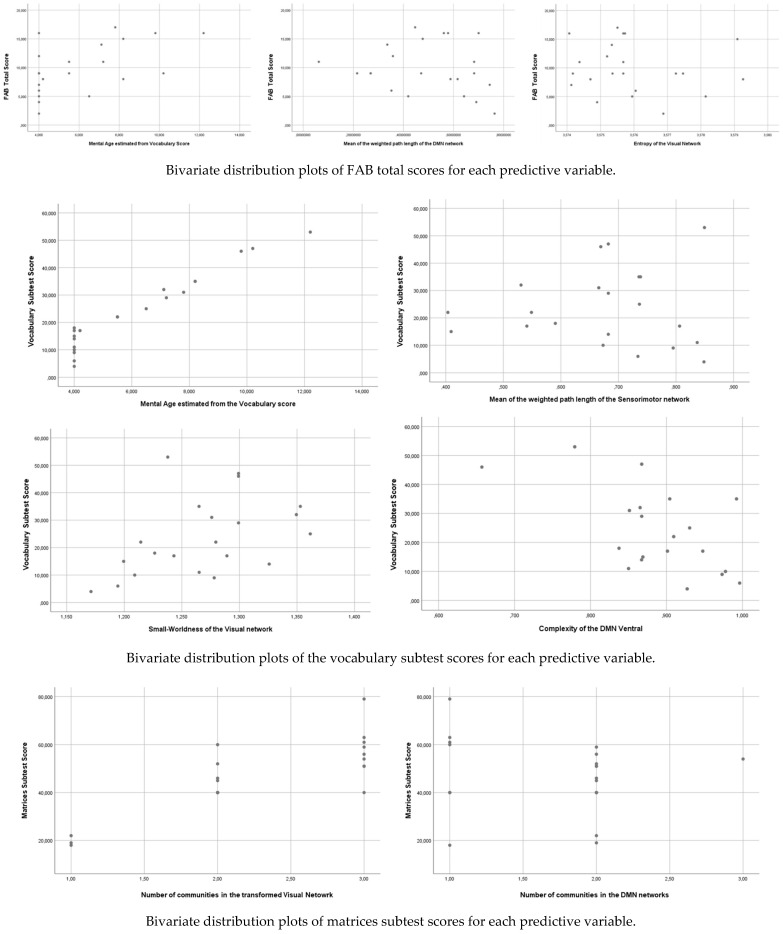
Bivariate plots representing the statistically significant effects of the linear model in predicting neuropsychological scores.

**Table 1 brainsci-11-00311-t001:** Relationships of Regions of Interest (ROIs) for the construction of the Default Mode Network (DMN) and subnetworks considered according to the AAL90 atlas.

DMN Partial		DMN Anterior		DMN Ventral		Sensorimotor		Visual	
Number in the AAL90 Atlas	Region Name	Number in the AAL90 Atlas	Region Name	Number in the AAL90 Atlas	Region Name	Number in the AAL90 Atlas	Region Name	Number in the AAL90 Atlas	Region Name
59	Parietal_Sup_L	29	Insula_L	35	Cingulum_Post_L	1	Precentral_L	43	Calcarine_L
60	Parietal_Sup_R	30	Insula_R	36	Cingulum_Post_R	2	Precentral_R	44	Calcarine_R
61	Parietal_Inf_L	31	Cingulum_Ant_L	37	Hippocampus_L	7	Frontal_Mid_L	45	Cuneus_L
62	Parietal_Inf_R	32	Cingulum_Ant_R	38	Hippocampus_R	8	Frontal_Mid_R	46	Cuneus_R
85	Temporal_Mid_L	87	Temporal_Pole_Mid_L	39	ParaHippocampal_L	19	Supp_Motor_Area_L	47	Lingual_L
86	Temporal_Mid_R	88	Temporal_Pole_Mid_R	40	ParaHippocampal_R	20	Supp_Motor_Area_R	48	Lingual_R
				55	Fusiform_L	57	PostcentralL	49	Occipital_Sup_L
				56	Fusiform_R	58	Postcentral_R	50	Occipital_Sup_R
				65	Angular_L	63	SupraMarginal_L	51	Occipital_Mid_L
				66	Angular_R	64	SupraMarginal_R	52	Occipital_Mid_R
				67	Precuneus_L	69	Paracentral_Lobule_L	53	Occipital_Inf_L
				68	Precuneus_R	70	Paracentral_Lobule_R	54	Occipital_Inf_R

**Table 2 brainsci-11-00311-t002:** List of estimated weighted indicators to determine the characteristics of each network analyzed.

	Description	Calculations
Functional Integration (FI)		
Number of communities	Number of independent communities detected in a group of specific ROIs. Estimated maximum number of statistically significant clusters in a random network.	
Mean of the path lengths	The path length of a node i (*Li*) is the average number of edges that must be crossed to go from node i to the remaining nodes in the network	$Li=\underset{i\epsilon N}{\sum }\left(\frac{1}{n-1}\underset{j\epsilon N,j\neq i}{\sum }dij\right)$ where *N* is the total number of nodes in the network, *n* is the number of nodes involved and *dij* is the shortest path length between node i and j.
Standard deviation of the path lengths	The characteristic path length is a global measure of the network, i.e., there is only one value for the entire network. It consists of the average path length of each node in the network.	$L=\frac{1}{N}\underset{i\epsilon N}{\sum }Li$
Functional Segregation (FS)		
Global clustering coefficient	This is the average value of the clustering coefficients, which is the fraction of triangles around a node, and is equivalent to the fraction of neighbors of the node that are neighbors among them.	$C=\frac{\sum \Gamma i}{\sum ki\left(ki-1\right)}$
Number of triangles	This is the number of connected triangles that can be estimated within a network in Euclidean space.	G=(V,E) An ordered pair in which V is a nonempty set of vertices and E is a set of edges. Where E consists of unordered pairs of vertices such as {x, y} E, then x and y are said to be adjacent.
Other measures		
Density	The network density (*D*) is the number of edges in the network in proportion to the total number of possible edges.	$D=\frac{K}{N\left(N-1\right)}$ where K is the number of edges in the network and *N* is the total number of nodes in the network.
Small world (Watts–Strogatz)	Networks that present a higher clustering coefficient than expected by chance and that, in addition, have a characteristic shortest path length.	$S=\frac{Cnorm}{Lnorm}=\frac{C/Crandom}{L/Lrandom}$ A network is said to represent this type of organization if the calculated index is greater than 1.
Complexity	The number of nodes and alternative paths that exist within a specific network	

**Table 3 brainsci-11-00311-t003:** Descriptions of the observed distributions of the criterion variables.

Criteria Variables	Mean (Standard Deviation)	Bootstrap 95% CI	Symmetry	Kurtosis
Mental Age Vocabulary	6.11 (2.51)	4.97–7.26	0.967	0.034
Mental Age Matrices	5.42 (1.53)	4.72–6.12	1.032	0.843
FAB (Frontal Assessment Battery) Score	9.62 (4.20)	7.71–11.53	0.215	0.681
Total Score Vocabulary	23.71 (13.91)	17.38–30.04	0.603	−0.452
Total Score Matrices	47.47 (13.76)	41.21–53.74	−0.063	0.879

**Table 4 brainsci-11-00311-t004:** Descriptions of the observed distributions of the complexity indicators in each group in the five subnetworks and the entire network that make up the DMN and their distribution according to the AAL atlas. The number of ROIs coincides with the description in Table 1. SD: standard deviation. DMN partial network (6 ROIs). DMN partial network (6 ROIs).

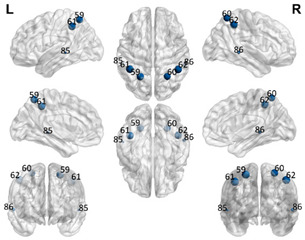	**Network Indicators**	**DS Group**		**Control Group**	
		**Mean**	**SD**	**Mean**	**SD**
	Number of communities	2.23	0.922	0.001	0.0001
	Mean of the weighted path	0.496	0.193	0.12	0.02
	Standard deviation of the weighted path	0.276	0.146	0.06	0.01
	Density	0.768	0.101	0.001	0.0001
	Small-worldness	1.027	0.088	0.0001	0.0001
	Global clustering coefficient	0.317	0.006	0.001	0.0001
	Complexity	0.822	0.238	0.11	0.02
	Segregation (triangles)	105.136	90.590	104.66	22.31
DMN Anterior (DMNa) partial network (6 ROIs)					
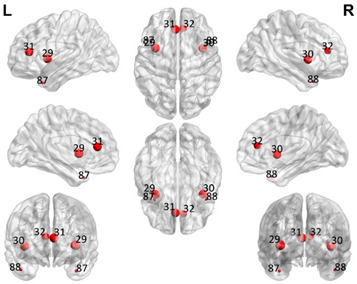	**Network Indicators**	**DS Group**		**Control Group**	
		**Mean**	**SD**	**Mean**	**SD**
	Number of communities	2.41	0.194	0.21	0.04
	Mean of the weighted path	0.423	0.043	0.10	0.02
	Standard deviation of the weighted path	0.299	0.027	0.05	0.01
	Density	0.768	0.021	0.0001	0.0001
	Small-worldness	1.027	0.018	0.0001	0.0001
	Global clustering coefficient	0.318	0.001	0.0001	0.0001
	Complexity	0.868	0.046	0.11	0.02
	Segregation (triangles)	109.181	18.035	103.14	21.98
DMN Ventral (DMNv) partial network (12 ROIs)					
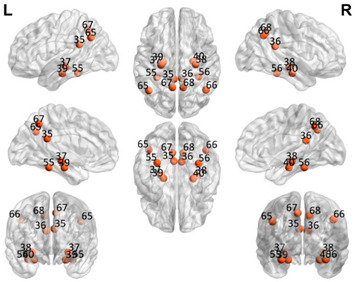	**Network Indicators**	**DS Group**		**Control Group**	
		**Mean**	**SD**	**Mean**	**SD**
	Number of communities	2.409	0.107	0.59	0.12
	Mean of the weighted path	0.605	0.033	0.08	0.01
	Standard deviation of the weighted path	0.232	0.029	0.03	0.01
	Density	0.454	0.001	0.001	0.001
	Small-worldness	1.240	0.014	0.05	0.01
	Global clustering coefficient	0.316	0.001	0.001	0.001
	Complexity	0.871	0.031	0.071	0.0001
	Segregation (triangles)	209.818	38.25	209.27	44.61
Sensorimotor (SM) partial network (12 ROIs)					
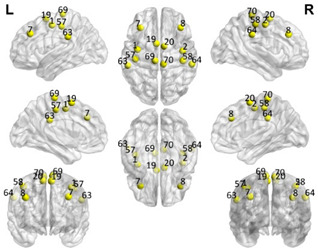	**Network Indicators**	**DS Group**		**Control Group**	
		**Mean**	**SD**	**Mean**	**SD**
	Number of communities	2.545	0.108	0.49	0.10
	Mean of the weighted path	0.677	0.027	0.08	0.01
	Standard deviation of the weighted path	0.202	0.032	0.22	0.04
	Density	0.454	0.001	0.001	0.001
	Small-worldness	1.230	0.013	0.07	0.01
	Global clustering coefficient	0.313	0.007	0.003	0.0008
	Complexity	0.884	0.027	0.06	0.001
	Segregation (triangles)	217.090	37.303	208.28	44.40
Visual (VIS) partial network (12 ROIs)					
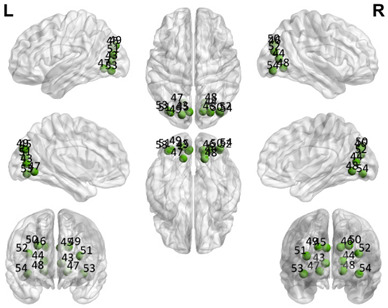	**Network Indicators**	**DS Group**		**Control Group**	
		**Mean**	**SD**	**Mean**	**SD**
	Number of communities	2.500	0.109	0.35	0.07
	Mean of the weighted path	0.783	0.016	0.07	0.01
	Standard deviation of the weighted path	0.115	0.009	0.04	0.009
	Density	0.454	0.001	0.0001	0.0001
	Small-worldness	1.271	0.011	0.06	0.001
	Global clustering coefficient	0.316	0.005	0.002	0.0006
	Complexity	0.930	0.011	0.04	0.008
	Segregation (triangles)	220.001	37.549	199.05	42.43
GLOBAL NETWORK ANALYSIS (48 ROIs)					
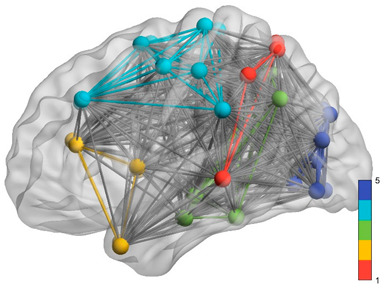	**Network Indicators**	**DS Group**		**Control Group**	
		**Mean**	**SD**	**Mean**	**SD**
	Number of communities	4.863	0.257	1.184	0.25
	Mean of the weighted path	0.690	0.022	0.07	0.01
	Standard deviation of the weighted path	0.248	0.030	0.08	0.01
	Density	0.152	0.020	0.001	0.0001
	Small-worldness	2.483	0.090	0.17	0.03
	Global clustering coefficient	0.312	0.001	0.001	0.0003
	Complexity	0.724	0.0617	0.03	0.008
	Segregation (triangles)	865,76	201.190	837.19	178.49

Purple: visual. Blue: sensorimotor. Green: ventral DMN (DMNv). Yellow: anterior DMN (DMNa). Red: DMN.

**Table 5 brainsci-11-00311-t005:** Pearson correlations between variables.

COMPLEXITY INDICATOR	DMN Partial			DMN Anterior			DMN Ventral			Sensorimotor			Visual			Observed Distribution		
	FAB	VOC	MAT	FAB	VOC	MAT	FAB	VOC.	MAT	FAB	VOC	MAT	FAB	VOC	MAT	FAB	VOC	MAT
Number of communities	0.221	0.333 **	0.572 **	−0.051	0.018	0.070	−0.026	0.110	0.328 **	−0.193 *	−0.238 **	0.375 **	0.172 *	0.247 **	0.510 **			
Mean of the path lengths	−0.365 **	0.184 *	−0.172 *	0.065	0.372 **	0.000	−0.241 **	−0.024	−0.275 **	−0.190	0.516 **	−0.127	−0.362 **	−0.435 *	−0.500 **			
SD of the path lengths	0.132	−0.164 *	0.002	−0.022	−0.188 *	−0.206 **	0.297 **	0.075	0.166 *	0.163 *	−0.047	0.001	0.376 **	0.644 **	0.491 **			
Density	−0.007	−0.341 **	−0.182 *	−0.007	−0.341 **	−0.182 *	0.011	0.007	0.022	0.003	0.012	0.008	0.084	0.026	0.003			
Small-world	0.007	0.341 **	0.182 *	0.007	0.341 **	0.182 *	−0.143 *	-0.003	−0.167 *	0.155	0.012	0.090	0.317 **	0.448*	0.197			
Global clustering coefficient	−0.105	0.022	0.033	0.017	−0.008	0.105	0.068	0.092	−0.311 *	0.131	−0.075	−0.107 *	0.245*	0.325 **	0.091			
Complexity	0.037	0.069	−0.008	0.042	−0.029	0.172 *	−0.175 *	−0.557 **	−0.139	−0.077	0.028	−0.267 *	−0.404 **	−0.436 **	−0.409 **			
Number of triangles	−0.181 *	−0.418	−0.457 *	−0.352 **	−0.427 **	−0.453 **	−0.359 **	−0.441 **	−0.417 **	−0.397 **	−0.438 *	−0.452 *	−0.368 **	−0.416	−0.438 **			
Mental age vocabulary																0.653 **	0.907 **	0.189 *
Mental age matrices																0.305 **	0.693 **	0.346 **

FAB = FAB total score; VOC = vocabulary subtest score; MAT = matrices subtest score. ** *p* < 0.001; * *p* < 0.05. In mental age vocabulary and matrices only the correlations with FAB, VOC and MAT were shown once.

**Table 6 brainsci-11-00311-t006:** Parameter estimation (*β_ij_*) for each of the criterion variables.

Criteria Variables	Predictor	Parameter	*p*	Effect Size		Observations
FAB total score	Mental age estimated from the vocabulary score	0.997	0.01	0.376	*AIC* = 124.367	Outliers: participant number 11 (Cook’s distance = 0.242)
	Mean of the weighted path length of the DMN network	−8.361	0.034	0.241		
Variables excluded	Step number 1: Number of communities in DMN partial; Number of triangles in the subnetworks DMN partial, DMN ventral, Sensoriomotor and Visual; SD of the path length of DMN ventral, Small-world in DMN ventral; Number of communities in Sensoriomotor network; mean and SD of the path lengths of Visual network; Small-world of the visual network, Global clustering coefficient of visual network; complexity of the visual network and Mental age derived from matrices subtest.					
Vocabulary subtest score	Mental age estimated from the vocabulary score	5.156	<0.001	0.950	*AIC* = 112.556	Outliers: participant number 16 (Cook’s distance = 0.388) and 19 (Cook’s distance = 0.264)
	Mean of the weighted path length of the sensorimotor network	−15.069	0.004	0.026		
	Small-worldness of the visual network	25.226	0.029	0.013		
	Complexity of the ventral DMN	−13.281	0.046	0.010		
Variables excluded	Step number 1: Number of communities, Mean and SD of the path lengths and number of triangles of DMN partial; Mean and SD of the path lengths, density and Small-world of the DMN anterior; Number of communities of DMN Ventral; Number of communities of Sensoriomotor and Visual networks. Step number 2: Density and Small-world of DMN partial; Number of triangles of DMN anterior; Complexity and Number of triangles of DMN ventral; Number of triangles of Sensoriomoto network a Mental Age derived from Matrices Test. Step number 3: Mean and SD of path lengths of Visual network; Global clustering coefficient, Complexity and Number of triangles of Visual network.					
Matrices subtest score	Number of communities in the visual network	14.581	0.004	0.562	*AIC* = 168.857	Outliers: participant number 7 (Cook’s distance = 0.432)
	Number of communities in the DMN networks	−24.149	0.042	0.247		
Variables excluded	Step number 1: Mean of the path lengths, Density, Small-world and Number of triangles of DMN partial; SD of path lengths, Density, Small-world, Complexity and Number of Triangles of DMN anterior; all the indicators (except Density) of the DMN ventral; Number of communities, Complexity and Number of triangles of Sensoriomotor network; Mean and SD of the path lengths, Small-world, Complexity and Number of triangles of Visual network. Step number 2: Mental age derived from vocabulary and matrices tests.					

No statistically significant differences were found in the observed distribution of complex indicators or any other variables (including the criteria variables) or sex.

## Data Availability

The data presented in this study are available on request from the corresponding author.

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
