# Peer review of "Complexity Analysis of the Default Mode Network Using Resting-State fMRI in Down Syndrome: Relationships Highlighted by a Neuropsychological Assessment"

_brainsci, 2021, doi:10.3390/brainsci11030311_

Round 1

Reviewer 1 Report

The authors have done a fair job revising the manuscript both structurally and linguistically. However, the main problem in the study design remains, namely that direct comparisons between controls and patients cannot in all fairness be trusted, especially on highly derirative features such as complexity of rs-fMRI subnetworks, given that the data was acquired in different scanners. 

Even if the manuscript were to focus on correlations with cognitive measures within the patient group, sample size is very marginal to permit reliable feature selection and estimation of true effects.

Reviewer 2 Report

Considering all of the limitations mentioned in the article (e.g., sample size, choice of test and approach to mental age estimation), it would be highly beneficial if the authors further continued their research in this area.

The data obtained in the current study are of great interest and have been correctly interpreted. The conclusions drawn be the authors are novel and can, in my opinion, further the understanding of DS.

On another note, I would like to mention that there is a small typo in line 134: as I understand, the control group was matched by AGE and sex, not by sex and sex.

Reviewer 3 Report

please see attachment.

Reviewer 4 Report

The authors investigated the relationship between the complexity indicators in the field of functional connectivity derived from rs-fMRI in Down syndrome samples and cognitive functional variables. In this presented study, 22 Down syndrome patients were assessed with KBIT and FAB, in comparison with 22 controls. The authors found that there was a significant difference in complexity indicators between groups: the control group showed less complexity than the Down syndrome group. 

This paper is well-written and the experiments are well-designed. I have two minor comments on this article.

  1. In Sec. 2.1., it would be better to provide a flowchart on the patient inclusion criteria and exclusion criteria, summarizing the patient's characteristics.
  2. It would be better to have at last one paragraph in Sec. 4 to discuss the limitation of the presented study.
  3. As shown in Table 4, it seems that the DS group has a larger SD than the control group. I am wondering if such a larger SD will affect the analysis.

Round 2

Reviewer 1 Report

The limitations of the study design (mainly small sample size) are acknowledged in the Discussion. However, the most serious, design, flaw of completing the rs-fMRI scans on the clinical sample from two scanners and using healthy control data from a third scanner, cannot be addressed post hoc.  

Author Response

This manuscript is a resubmission of an earlier submission. The following is a list of the peer review reports and author responses from that submission.

Round 1

Reviewer 1 Report

Given the known impact of magnet specs and acquisition parameters on the signal and derived rs-fMRI indices it is crucial that the authors compare the acquisition parameters used on the patient and control groups. If these features are not identical between patient and control groups, direct comparisons are untenable. In that case the results should be limited to correlations with cognitive scores in the patient group

Computing graph metrics separately for each subnetwork is somewhat counterintuitive. The underlying principle of graph theory network modeling is to explore network structure based on the data and taking into account individual differences. Also in this manner the number of potential correlates of cognitive test scores increases dramatically (I counted 54 correlated metrics). Stepwise regression is notorious for overfiting problems especially when colinearity is present. WLS unfortunately is not a safe remedy for the small sample size (N=22) relative to the number of estimated parameters. 

Author Response

REV Given the known impact of magnet specs and acquisition parameters on the signal and derived rs-fMRI indices it is crucial that the authors compare the acquisition parameters used on the patient and control groups. If these features are not identical between patient and control groups, direct comparisons are untenable. In that case, the results should be limited to correlations with cognitive scores in the patient group.

We include a specific sentence for this question to clarify this situation.

Computing graph metrics separately for each subnetwork is somewhat counterintuitive. The underlying principle of graph theory network modeling is to explore network structure based on the data and taking into account individual differences.

We understand the reviewer's point of view, but it should be considered that in the case of the DMN there is ample evidence about the behavior of the subnets of that structure. If it is desired, as in our case, to maintain a certain neurofunctional reference, it is essential to respect the subnets that define specific neurofunctional functioning.

 Also, in this manner the number of potential correlates of cognitive test scores increases dramatically (I counted 54 correlated metrics). Stepwise regression is notorious for overfiting problems especially when colinearity is present. WLS unfortunately is not a safe remedy for the small sample size (N=22) relative to the number of estimated parameters.

Indeed, regression models have a limited role given the conditions under which they have been estimated. The use of estimates of the TSLS type or similar was ruled out since the effects on the improvement of tolerance and the behavior of the residuals did not advise it. Estimates such as WLS have a certain advantage in small samples since it reduces the effects of alterations in the distributions (as in our case) and the effects of outlier values. To clarify the way in which the three regression models have been proposed, we have included a sentence that clarifies this detail, since the reviewer is right to make this consideration. In each model, only the variables with statistically significant correlation with each criterion variable were used as regressors. In this way the number of regressors was much lower and neither the order nor the saturation condition of the linear model was compromised.

Reviewer 2 Report

This study used fMRI to study the default mode network (DMN) in brains of individuals with down syndrome (DS). A control group was also included. Results of the study showed that the brains of individuals with DS had more activity and variability in the DFM than controls. Although this is a valid study, with noteworthy results, it needs heavy editing and proof-reading. Relatedly, please note typographical errors throughout text. A lot of sentences are whole paragraphs, which makes for a tiring read. I strongly suggest to heavily revise the manuscript to make it comprehensible, sharp and clear. A few notes below:

Abstract: Define KBIT and FAB tests in the abstract.

Introduction: In the Introduction line 42, the abbreviation DTI is noted to be Diffusion Stress. Please double this check this (Do the authors mean diffusion tensor imaging?).

The introduction walks the reader through neuroimaging techniques in some detail. The applicability of fMRI and its ability to analyze brain networks is also given. The default mode network (DMN), and how its patterns vary across patient populations is then introduced. Specifically, differences on its function and connectivity across patients with Alzheimer’s disease, Down Syndrome, and controls are noted. The structure on starting broad and narrowing down to the specific purposes of the paper are a useful means to ease into the topic.

A useful exercise would include going through the manuscript and shorten sentences. Most of them are too verbose to the point that they become non-sensical, for example lines 94- 98 is a paragraph. It is also one sentence. Whatever is written after “visual DMN networks” is incoherent.  

In the introduction, it would be useful to give a one-sentence introduction to “g-factor” scores.

Lines 58-60 of the introduction – it is unclear what groups are being spoken about.

Typically, regions within the limbic system are associated with AD. Thus, elaboration on how “impaired development of the prefrontal cortex” (Lines 90 – 92 of Introduction)  may be an indication of AD, is required.

The sentence “fewer complex structures are associated with better health” (lines 113-114) needs to be rewritten or clarified.

Line 124 delete phrase “Because of the above…”

Line 128, typically a framework would constitute a theoretical background rather than cognitive domains. Either include a theoretical background or re-word.

The objective of the study is poorly worded. It would be possible to substitute paragraph-long sentences for shorter ones, and state the hypotheses in a separate sentence.

Materials and methods

This section was long and wordy, making it hard to follow through. I suggest some editing to the whole section and to present only information relevant to this particular study. A few additional notes:

What do the authors mean by “accidental” sampling?

How was degree for intellectual disability classified? A list of cut-offs for levels of intellectual disability should be noted.

In line 153 the authors mention a second control group, but there seems to be just one control groups, and why would they want two?

Line 26, what is the KBIT test? The abbreviation should be spelled out, and the test should be described.

Results

A demographic table should be presented.

What do authors mean by “importance” in Table 5?

Discussion

Line 140. What is “extraordinarily large”? Significantly large?

Conclusions

“This is adjusted to our hypothesis based in the complexity structure of brain connectivity as a biomarker of healthy status does not make sense.” Does not make sense

“There is an enormous change in the behaviour of networks between subjects.” Change was not studied in this paper.

Please revisit and revise – it may be an excellent study, but it’s a tough read.

Author Response

Abstract: Define KBIT and FAB tests in the abstract.

Done

Introduction: In the Introduction line 42, the abbreviation DTI is noted to be Diffusion Stress. Please double this check this (Do the authors mean diffusion tensor imaging?).

Done and error solved.

The introduction walks the reader through neuroimaging techniques in some detail. The applicability of fMRI and its ability to analyze brain networks is also given. The default mode network (DMN), and how its patterns vary across patient populations is then introduced. Specifically, differences on its function and connectivity across patients with Alzheimer’s disease, Down Syndrome, and controls are noted. The structure on starting broad and narrowing down to the specific purposes of the paper are a useful means to ease into the topic.

A useful exercise would include going through the manuscript and shorten sentences. Most of them are too verbose to the point that they become non-sensical, for example lines 94- 98 is a paragraph. It is also one sentence. Whatever is written after “visual DMN networks” is incoherent.

In the introduction, it would be useful to give a one-sentence introduction to “g-factor” scores.

Lines 58-60 of the introduction – it is unclear what groups are being spoken about.

 Typically, regions within the limbic system are associated with AD. Thus, elaboration on how “impaired development of the prefrontal cortex” (Lines 90 – 92 of Introduction) may be an indication of AD, is required.

The sentence “fewer complex structures are associated with better health” (lines 113-114) needs to be rewritten or clarified.

Line 124 delete phrase “Because of the above…”

Line 128, typically a framework would constitute a theoretical background rather than cognitive domains. Either include a theoretical background or re-word.

The objective of the study is poorly worded. It would be possible to substitute paragraph-long sentences for shorter ones and state the hypotheses in a separate sentence.

We have included new phrases to solve the question that the reviewer asks us and make the introduction easier.

Materials and methods

This section was long and wordy, making it hard to follow through. I suggest some editing to the whole section and to present only information relevant to this particular study. A few additional notes:

What do the authors mean by “accidental” sampling?

Clarified (non random)

How was degree for intellectual disability classified? A list of cut-offs for levels of intellectual disability should be noted.

Clarified in the paper. 

In line 153 the authors mention a second control group, but there seems to be just one control groups, and why would they want two?

Solved

Line 26, what is the KBIT test? The abbreviation should be spelled out, and the test should be described.

Done

Results

A demographic table should be presented.

In this case, we do not consider it necessary to include one more table since there is no more demographic information than the one shown in the draft.

What do authors mean by “importance” in Table 5?

Solved

Discussion

Line 140. What is “extraordinarily large”? Significantly large?

We have clarified the details you request

Conclusions

“This is adjusted to our hypothesis based in the complexity structure of brain connectivity as a biomarker of healthy status does not make sense.” Does not make sense

We modify this concept.

“There is an enormous change in the behaviour of networks between subjects.” Change was not studied in this paper.

I agree with the reviewer. We have modified the sentence

Round 2

Reviewer 1 Report

One of the main goals of this report is to identify potentially aberrant complexity within the visual and within each of two DMN subnetworks in adolescents and young adults with Down Syndrome as compared to unimpaired controls. However, not only the clinical group was tested on different scanners (same model does not necessarily imply same hardware [e.g. channels], field homogeneity, and noise levels), but the control groups was tested on more than one scanner of unspecified type (data came from the connectome project; different scanner and ). Therefore, for all practical purposes direct comparisons between the two groups is not acceptable. 

There may be some value in the observed associations between graph metrics and cognitive test scores. However to properly evaluate and interpret those, a table of zero-order correlations would be very helpful. Also simple bivariate plots between for instance FAB/KBIT and weighted path length in the SMN would help visualize the reportedly peculiar distributions.

Reviewer 2 Report

please see below.